

# Extending Square Conservation to Arbitrarily Structured C-grids with Shallow Water Equations

Lilong Zhou[1,3], Jinming Feng[2], Lijuan Hua[1]

[1]College of Earth and Planetary Sciences, University of Chinese Academy of Sciences
[2]Key Laboratory of Regional Climate-Environment for Temperate East Asia, Institute of Atmospheric Physics, Chinese Academy of Sciences
[3]Numerical Weather Prediction Center of China Meteorological Administration

*Correspondence to*: Jinming Feng (fengjm@tea.ac.cn)

**Abstract.** The square conservation theory is widely used on latitude–longitude grids, but it is rarely implemented on quasi-uniform grids, given the difficulty involved in constructing anti-symmetrical spatial discrete operators on these grids. Increasingly more models are developed on quasi-uniform grids, such as arbitrarily structured C-grids. Thuburn–Ringler–Skamarock–Klemp (TRiSK) is a shallow water dynamic core on an arbitrarily structured C-grid. The spatial discrete operator of TRiSK is able to naturally maintain the conservation properties of total mass, total absolute vorticity and instantaneous total

energy. The first 2 integral invariants are entirely conserved during integration, but the total energy dissipates when using the dissipative temporal integration schemes, i.e., Runge-Kutta. The method of strictly conserving the total energy simultaneously uses both an anti-symmetrical spatial discrete operator and square conservative temporal integration scheme. In this study, we demonstrate that square conservation is equivalent to energy conservation in both a continuous shallow water system and a discrete shallow water system of TRiSK, attempting to extend the square conservation theory to the TRiSK framework. To

overcome the challenge of constructing an anti-symmetrical spatial discrete operator, we unify the unit of evolution variables of shallow water equations by Institute of Atmospheric Physics (IAP) transformation, expressing the temporal trend of the evolution variable by using the original operators of TRiSK. Using the square conservative Runge-Kutta scheme, the total energy is completely conserved, and there is no influence on the properties of conserving total mass and total absolute vorticity. In the standard shallow water numerical test, the square conservative scheme not only helps maintain total conservation of the

three integral invariants but also creates less simulation error norms.

## 1 Introduction

The maintenance of integral constraints is necessary to determine the true solution, following a path upon which the statistics are analogous to those of the true solution (Arakawa, 1977). Shallow water equation sets, without any outer sources and frictions, have five basic physical conservation properties, including total mass, total energy, total absolute vorticity (total

potential vorticity), total potential enstrophy and total angular momentum. These conservation properties are important in an





atmosphere model, especially with regard to long-term simulation; however, in a discrete system, some conservation properties cannot be maintained (Wang, 2008).

A numerical scheme, with an energy conservation or energy dissipative property, is prerequisite to prevent nonlinear computational instability; however, an energy dissipative scheme will limit short-waves, which is meaningful for the
atmosphere (Shen, 2013; Zeng, 1981).

On a latitude–longitude grid, energy is able to be entirely conserved by constructing a square conservative finite difference scheme (Ji and Wang, 1991), or a multi-conservation finite difference scheme (Wang and Ji, 2003), the former of which is better developed. Wang and Ji (1994a) discussed the square conservative scheme (SCS), the complete square conservative scheme (CSCS), the instantaneous square conservative scheme (ISCS) and the explicit complete square conservative scheme
with adjustable time intervals (ECSCSA). The ISCS maintains square conservation only in the spatial discrete scheme and not in the temporal integration scheme, which implies the spatial discrete operator of the model is a square conservative (i.e., an anti-symmetrical operator). However, the temporal integration scheme does not possess the square conservation property because the model is energy dissipative during integration. The CSCS maintains square conservation in both the spatial and temporal schemes. The model, which adopts CSCS, is able to maintain complete energy conservation during integration. The
first step of applying the square conservation theory is to construct an anti-symmetrical spatial discrete operator and then integrate the model with a square conservative temporal integration scheme, i.e., a modified predict-corrector, modified leap-frog (Wang and Ji, 2006), harmonious dissipative operators (Wang and Ji, 1994b), etc.

To improve integration precision on the temporal direction of the square conservative scheme, a new class of Runge-Kutta schemes, hereafter CRK, were developed by Wang et al. (1996). The CRK scheme maintains the complete square conservation
property by adjusting the length of temporal integration steps and maintaining the same integral precision order as the original Runge-Kutta scheme, hereafter RK.

The SCS was implemented in the grid-point atmospheric model of IAP LASG (GAMIL, Wang et al. 2004, Wang and Ji, 2006). GAMIL is widely used in climate simulation (Li et al., 2007, 2013; Wu and Li, 2008). The square conservation theory is rarely used on quasi-uniform grids or nonuniform grids because it is hard to construct a spatial discrete operator with an anti-
symmetrical property on those grids.

In the most recent two decades, to avoid the polar problem of the latitude–longitude grid, increasingly more atmosphere models have been built on the quasi-uniform grid, i.e., spectral transform methods (Swarztrauber, 1996); the finite volume method (Lin, 2004; Putman and Lin, 2007; Chen and Xiao, 2008); and an extension on the finite difference method to the generalized curvilinear coordinates (Toy and Nair, 2017).

Thuburn et al. (2009) and Ringler et al. (2010), provided a spatial discrete scheme based on arbitrarily structured C-grids, known as Thuburn–Ringler–Skamarock–Klemp (TRiSK). TRiSK is able to conserve the total mass and total absolute vorticity, and the total energy is instantaneously conserved. These important properties enable models using quasi-uniform Voronoi grids, the accuracies of which are similar to latitude–longitude grids (Weller et al., 2012). Based on Thuburn et al. (2009) and



Ringler et al. (2010), a global/regional model, the Model for Prediction Across Scales (MPAS), was developed by the National Center for Atmospheric Research (NCAR) and Los Alamos National Laboratory (LANL) (Skamarock et al., 2012, 2018). Although the semi-discrete operator designed by Ringler et al. (2010) results in instantaneous energy conservation, the total energy is still dissipative while using dissipative temporal integration schemes, i.e., Runge-Kutta. Energy will be completely

conserved only when the spatial discrete operator is anti-symmetrical and the temporal integration scheme is square conservative (Wang and Ji, 2006).

The main obstacle of extending square conservation to the quasi-uniform grids is constructing the anti-symmetrical spatial discrete operator. Because many quasi-uniform grids are unstructured and the shapes of cells are not uniform, it is difficult to find the next or previous cell. In this study, we use the instantaneous energy conservation property of TRiSK to overcome the

challenge of constructing an anti-symmetrical spatial operator on a quasi-uniform grid. After using CRK as a temporal integration scheme, the square conservative constrains are satisfied for both spatial and temporal directions, and the total energy, total mass and total absolute vorticity are completely conserved during the integration.

This paper is presented as follows: In section 2, we review the TRiSK framework which was described by Ringler et al. (2010). Section 3 describes the method of extending square conservation to TRiSK in 3 parts. The first part presents a review of square

conservation and a demonstration of the equivalent relationship between square conservation and energy conservation in a continuous shallow water system. In the second part of section 3, we obtain the anti-symmetrical spatial discrete operator by using the derivative rule and the energy conservation property of TRiSK, a method that is key to extending square conservation to TRiSK. In the last part of section 3, we review a new type of Runge-Kutta with 4th-order precision, which was developed by Wang et al. (1996) as the square conservative temporal integration scheme. In section 4, by using the square conservation

scheme, we demonstrate that the total mass and total absolute vorticity remain perfectly conservative. Section 5 exhibits the results of three different numerical tests, including the 2nd, 5th and 6th test cases mentioned by Williamson (1992).

## 2 Introduction

The shallow water equation set may be written in a flux format as follows:

$$\frac{\partial \boldsymbol{u}}{\partial t} - \xi_a \boldsymbol{k} \times \boldsymbol{u} + \nabla E = 0 , \tag{1}$$

$$\frac{\partial \phi}{\partial t} + \nabla \cdot (\phi \boldsymbol{u}) = 0 , \tag{2}$$

where, $\xi_a = \xi + f$ denotes the absolute vorticity; $\xi = \nabla \times \boldsymbol{u}$ represents the relative vorticity; $f = 2\Omega \sin \theta$ is the Coriolis parameter; $E = K + g(h + h_s) = K + \phi + \phi_s$, $\phi = gh$ is the geopotential depth of the fluid; $\phi_s = gh_s$ is the geopotential height of the underlying surface; $\phi_t = \phi + \phi_s$ is the free surface (top) of the fluid; $K = \frac{|\boldsymbol{u}|^2}{2}$ is the kinetic energy; $\boldsymbol{u}$ is the velocity vector; $h$ and $h_s$ are the fluid thickness and surface height, respectively; $\theta$ represents the latitude; and $g$ and $\Omega$ are

acceleration of gravity and angular velocity of the earth.



In Ringler et al. (2010), the total energy is defined as

$$\epsilon_{R10} = hK + gh\left(\frac{1}{2}h + h_s\right)$$

To simplify the derivation in the following context, we define the total energy as

$$\epsilon = g\epsilon_{R10} = \phi K + \frac{1}{2}\phi^2 + \phi\phi_s = \|\phi K\| + \left\|\frac{1}{2}\phi^2\right\| + \|\phi\phi_s\| , \tag{3}$$

where $\|\cdot\| = \sqrt{(\cdot,\cdot)}$ denotes the 2-norm. The inner product $(\cdot,\cdot)$ is defined as

$$(X,Y) = \oiint_{\Omega} X \cdot Y \, ds$$

where $\Omega$ is the entire spherical surface.

**Figure 1.** Definition of elements in a discrete system. Blue arrows represent the indicator function $\boldsymbol{n_{e,i}}$ and red arrows are the indicator
function $\boldsymbol{t_{e,v}}$





Per the description provided in Ringler et al. (2010), velocity points are on the edges of each cell, the mass and kinetic energy points are in the center of the cell and vorticity points are on the vertices of the cell. The shallow water equation set may be expressed as a semi-discrete form:

$$\frac{\partial u_e}{\partial t} - Q_e^\perp + [\nabla E]_e = 0 ,$$ (4)

$$\frac{\partial \phi_i}{\partial t} + [\nabla \cdot (\phi u)]_i = 0 ,$$ (5)

where $u, v$ are the normal velocity and tangent velocity. The subscript $e$ signifies that the variable is defined on edge; the subscript $i$ signifies that the variable is defined at the center of a cell. $Q_e^\perp$ is the absolute vorticity flux on the tangent direction $\perp$ of the edge $e$, which is computed according to Eq. (49) in Ringler et al. (2010).

$$[\nabla E]_e = \frac{1}{d_e} \sum_{i \in CE(e)} -n_{e,i} E_i$$

$$[\nabla \cdot (\phi u)]_i = \frac{1}{A_i} \sum_{e \in EC(i)} n_{e,i} l_e \phi_e u_e$$

where $n_{e,i}$ is an indicator function, defined as $n_{e,i} = 1$ when $n_e$ is an outward normal vector of cell $i$, and $n_{e,i} = -1$ when $n_e$ is an inward normal vector of cell $i$; $l_e$ is the length of edge $e$; $i \in CE(e)$ denotes the two cells that share edge $e$; and $e \in EC(i)$ is the set of edges that define the boundary of cell $i$. The potential vorticity on edge $q_e$ may be computed by the midpoint method (Ringler et al. (2010), Eq. (50)) or the linear interpolation method (Weller, 2012, Eq. (5)). The details are presented in
Figure 1.

### 3 Extending the square conservation to TRiSK

As mentioned in section 1, to obtain the complete square conservation property, the spatial discrete operator must be anti-symmetrical, and the temporal integration scheme is square conservative. Therefore, in this section, the method of extending the square conservation to TRiSK is introduced in three parts. Subsection 3.1 reviews the concept of square conservation,
demonstrating the equivalent relationship between the square conservation and energy conservation. Subsection 3.2 constructs the anti-symmetrical spatial discrete operator. Subsection 3.3 introduces the square conservative temporal integration scheme by reviewing a new type of Runge-Kutta (CRK), which was developed by Wang et al. (1996).

### 3.1 Relationship between Square Conservation and Energy Conservation

First, we review the concept of anti-symmetrical operators and square conservation according to the study of Wang et al.
(1996), considering the nonlinear evolution equation in operator form:

$$\frac{\partial \boldsymbol{F}}{\partial t} + \mathcal{L}\boldsymbol{F} = 0 ,$$ (6)





*Definition. Suppose H is a complete inner product space on R and $\mathcal{L}$ is an $H \to H$ operator; if $\mathcal{L}$ satisfies the following inner product equation*

$$(\mathcal{L}\boldsymbol{F}, \boldsymbol{F}) = 0 \,, \tag{7}$$

*then $\mathcal{L}$ is termed an anti-symmetrical operator.*

The result of multiplying $F$ on both sides of (6) and integrating globally is the square conservation property:

$$\frac{d}{dt}\|\boldsymbol{F}\|^2 = 0 \,, \tag{8}$$

Next, we begin determine the relationship between energy conservation and square conservation. In the TRiSK framework, the evolution variables are $\boldsymbol{u}$ and $\phi$.

The unified unit of evolution variables is the prerequisite of constructing the square conservation system. The evolution

variables are unified by IAP transformation, and the original evolution variable $\boldsymbol{u}$ is replaced by the new evolution variable $\boldsymbol{U} = \sqrt{\phi}\boldsymbol{u}$, after completing IAP transformation.

$$\boldsymbol{F} = \begin{pmatrix} \boldsymbol{U} \\ \phi \end{pmatrix} = \begin{pmatrix} \sqrt{\phi}\boldsymbol{u} \\ \phi \end{pmatrix}, \tag{9}$$

The physical significance of $\sqrt{\phi}$ is the phase speed of the external-gravity wave, and the shallow water equation set may be rewritten as a vector format:

$$\frac{\partial \boldsymbol{F}}{\partial t} + \mathcal{L}\boldsymbol{F} = \frac{\partial}{\partial t}\begin{pmatrix} \boldsymbol{U} \\ \phi \end{pmatrix} + \mathcal{L}\begin{pmatrix} \boldsymbol{U} \\ \phi \end{pmatrix} = 0 \,, \tag{10}$$

As defined in section 2, $\phi_t = \phi + \phi_s$

$$\frac{\partial \phi_t}{\partial t} = \frac{\partial \phi}{\partial t} + \frac{\partial \phi_s}{\partial t}$$

The surface height is determined to be independent of time,

$$\frac{\partial \phi_s}{\partial t} = 0$$

Therefore,

$$\frac{\partial \phi_t}{\partial t} = \frac{\partial \phi}{\partial t} \,, \tag{11}$$

Defining $\boldsymbol{F}_t = \begin{pmatrix} \boldsymbol{U} \\ \phi_t \end{pmatrix}$, according to (9) and (11), we have

$$\frac{\partial \boldsymbol{F}_t}{\partial t} = \frac{\partial \boldsymbol{F}}{\partial t} \,, \tag{12}$$

Multiplying (12) by $\boldsymbol{F}_t$ on both sides, and integrating globally





$$\frac{d}{dt}\left\|\frac{1}{2}F_t^2\right\| = \left(F_t, \frac{\partial F}{\partial t}\right)$$

$$= \oiint_\Omega U\frac{\partial U}{\partial t} + (\phi + \phi_s)\frac{\partial \phi}{\partial t}\, ds$$

$$= \oiint_\Omega \frac{\partial}{\partial t}\left(\frac{1}{2}|U|^2\right) + \frac{\partial}{\partial t}\left(\frac{1}{2}\phi^2 + \phi\phi_s\right) ds$$

$$= \frac{d}{dt}\left(\|\phi K\| + \left\|\frac{1}{2}\phi^2\right\| + \|\phi\phi_s\|\right)$$

$$= \frac{d\epsilon}{dt} = 0$$

Accordingly, square conservation is equivalent to energy conservation in a continuous system.

### 3.2 Constructing the anti-symmetrical spatial discrete operator

Assuming a continuous-in-time system, the evolution equation of $U$ is able to be expressed as

$$\frac{\partial U}{\partial t} = \sqrt{\phi}\frac{\partial u}{\partial t} + \frac{u}{2\sqrt{\phi}}\frac{\partial \phi}{\partial t}, \qquad (13)$$

This formula is key to connecting square conservation and energy conservation; it is difficult to directly construct the anti-symmetrical operator on quasi-uniform grids.

*Theorem. The operators $\mathcal{M}$ and $\mathcal{N}$ satisfy*

$$\begin{cases} \frac{\partial u}{\partial t} + \mathcal{M}u = 0 \\ \frac{\partial \phi}{\partial t} + \mathcal{N}\phi = 0 \end{cases}, \qquad (14)$$

*and*

$$(\mathcal{M}u, \phi u) + (\mathcal{N}\phi, E) = 0$$

*After IAP transformation (9), the evolution equation of $U$ may be expressed as (13), and (14) may be rewritten as (10).*

*If the operator $\mathcal{L}$ satisfies (10), then $\mathcal{L}$ is an anti-symmetrical operator.*

*Proof.*

$$\frac{\partial U}{\partial t} = \sqrt{\phi}\frac{\partial u}{\partial t} + \frac{u}{2\sqrt{\phi}}\frac{\partial \phi}{\partial t} = -\sqrt{\phi}\mathcal{M}u - \frac{u}{2\sqrt{\phi}}\mathcal{N}\phi$$

$$(\mathcal{L}F, F) = \left(\frac{\partial U}{\partial t}, U\right) + \left(\frac{\partial \phi}{\partial t}, \phi\right)$$

$$= \oiint_\Omega U\frac{\partial U}{\partial t} + \phi\frac{\partial \phi}{\partial t}\, ds$$

$$= \oiint_\Omega U\left(-\sqrt{\phi}\mathcal{M}u - \frac{u}{2\sqrt{\phi}}\mathcal{N}\phi\right) - \phi\mathcal{N}\phi\, ds$$

$$= \oiint_\Omega -\phi u \cdot \mathcal{M}u - \frac{|u|^2}{2}\mathcal{N}\phi - \phi\mathcal{N}\phi\, ds$$





$$= -(\mathcal{M}u, \phi u) - (\mathcal{N}\phi, E)$$

$$= 0$$

This theorem is proved in a continuous system, but the model is built in a discrete system; therefore, it is necessary to discuss the situation in a discrete system.

Following Ringler et al. (2010), we set the discrete operators $M$ and $N$ as:

$$Mu = [\nabla E]_e - Q_e^\perp$$

$$N\phi = [\nabla \cdot (\phi u)]_i$$

And the semi-discrete shallow water equation set becomes

$$\frac{\partial u}{\partial t} + Mu = 0 \ , \tag{15}$$

$$\frac{\partial \phi}{\partial t} + N\phi = 0 \ , \tag{16}$$

Because the semi-discrete operator of TRiSK has an instantaneous energy conservation property, it is easy to prove $(Mu, \phi u) + (N\phi, E) = 0$. (Details in Ringler et al. (2010), section 3.7.2)

There are cells, edges and vertices presented as three types of points on a spherical centroidal Voronoi tessellation (SCVT) grid, which is the mesh used by TRiSK. We define the inner product for different types of points as:

$$(X, Y)_i = \sum_{i=1}^{nCells} X_i \cdot Y_i \cdot A_i$$

$$(X, Y)_e = \sum_{e=1}^{nEdges} X_e \cdot Y_e \cdot A_e$$

where $X_i, Y_i$ are the variables in the cell; $X_e, Y_e$ denote any variables on the edge; $A_i, A_e$ are the areas for each cell and edge; $A_e = d_e \times l_e$, $d_e$ is the distance between the two cells' centers on edge $e$; $l_e$ is the length of edge $e$; $nCells$ denotes the total cell number; and $nEdges$ is the total edge number.

$$\left(Mu, (\widehat{\phi})_e u\right)_e + (N\phi, E)_i = 0 \ , \tag{17}$$

Combining (10) and (13), and rewriting into a discrete system

$$\frac{\partial \boldsymbol{F}}{\partial t} + L\boldsymbol{F} = \frac{\partial}{\partial t}\begin{pmatrix} U_e \\ \phi_i \end{pmatrix} + L\begin{pmatrix} U_e \\ \phi_i \end{pmatrix} = 0 \ , \tag{18}$$

$$\frac{\partial U_e}{\partial t} = \sqrt{\phi_e}\frac{\partial u_e}{\partial t} + \frac{u_e}{2\sqrt{\phi_e}}\frac{\partial \phi_e}{\partial t} = -\sqrt{\phi_e}Mu - \frac{u_e}{2\sqrt{\phi_e}}N\phi \ , \tag{19}$$

As shown in the Appendix A, we have the discrete anti-symmetrical operator $L$

$$(L\boldsymbol{F}, \boldsymbol{F})_d = \left(U, \frac{\partial U}{\partial t}\right)_e + \left(\phi, \frac{\partial \phi}{\partial t}\right)_i = 0 \ , \tag{20}$$

The subscript $d$ represents that the inner product is computed in a discrete system.

Thus, the discrete evolution equation set becomes





$$\frac{\partial U_e}{\partial t} + \sqrt{\phi_e}Mu + \frac{u_e}{2\sqrt{\phi_e}}N\phi = 0 \ , \tag{21}$$

$$\frac{\partial \phi_i}{\partial t} + N\phi = 0 \ , \tag{22}$$

The model will be instantaneous square conservative by incorporating (21) and (22) as the evolution equation set.

### 3.3 Constructing the temporal integration scheme with the square conservation property

The model is integrated in a discrete-in-time system, for the sake of guaranteeing complete square conservation, a square conservative temporal integration scheme is necessary. As CRK has the advantage of maintaining complete square conservation with a high order of integral precision, the 4[th]-order CRK scheme in TRiSK is adopted to obtain high integral precision and a long-time step. To completely introduce the square conservative temporal integration scheme, we review some of the details in Wang et al. (1996).

The 4[th]-order CRK may be expressed as

$$F^{n+1} = F^n + \tau_n \varphi(F^n, \tau) \ , \tag{23}$$

where $\tau_n$ is an adjustable time step and $\tau$ is the integral time step of the model.

$$\varphi(F^n, \tau) = \tau \frac{R_1 + 2R_2 + 2R_3 + R_4}{6}$$

$$\begin{cases} R_1 = -LF^n \\ R_2 = -L\left(F^n + \frac{1}{2}\tau R_1\right) \\ R_3 = -L\left(F^n + \frac{1}{2}\tau R_2\right) \\ R_4 = -L(F^n + \tau R_3) \end{cases}$$

Taking square operators on both sides of (23), delineating $\varphi^n = \varphi(F^n, \tau)$

$$\|F^{n+1}\|^2 = \|F^n\|^2 + 2\tau_n(\varphi^n, F^n) + \tau_n{}^2\|\varphi^n\|^2 \ , \tag{24}$$

We notice that although the spatial discrete operator $L$ is anti-symmetrical, the total energy at the $n + 1$ time point remains different from that at the $n$ time point. Energy is able to be completely conserved by satisfying the following equation:

$$\|F^{n+1}\|^2 = \|F^n\|^2$$

Therefore,

$$2\tau_n(\varphi^n, F^n) + \tau_n{}^2\|\varphi^n\|^2 = 0$$

$$\tau_n = -\frac{2(\varphi^n, F^n)}{\|\varphi^n\|^2}$$

Considering the fitness when $\tau \to 0$, as described in Eqs. (17)–(18) in Wang et al. (1996)

$$\tau_n = \frac{\tau}{3\|\varphi^n\|^2}[(R_1, R_2) + (R_2, R_3) + (R_3, R_4)] \ , \tag{25}$$





Once adopting the CRK scheme as the temporal integration scheme, the model will be completely square conservative, which implies the total energy will be completely conserved from the beginning to the end of the integration. The CRK scheme is expected to perform better than RK in a numerical test. Moreover, CRK decays to RK by setting $\tau_n = \tau$.

While the integral time step is modified from $\tau$ to $\tau_n$, the precision order of CRK is the same as RK, when constructing CRK

based on the $n$th order RK, CRK has $n$th order precision either, a conclusion proven by Theorem 1 in Wang et al. (1996).

**4 Mass and Absolute Vorticity Conservation**

In the CSCS introduced above, although the integral time step is modified from $\tau$ to $\tau_n$, the total mass and total absolute vorticity are nevertheless conserved. In the following demonstrations, we notice that the mass conservation property and absolute vorticity conservation property are independent of temporal integration.

**4.1 Mass Conservation**

Considering the total mass, multiplying (5) by $A_i$ and summing all cells,

$$\sum_{i=1}^{nCells} A_i \frac{\partial \phi_i}{\partial t} = -\sum_{i=1}^{nCells} A_i [\nabla \cdot (\phi u)]_i = -\sum_{i=1}^{nCells} \sum_{e \in EC(i)} n_{e,i} l_e \phi_e u_e = -\sum_{e=1}^{nEdges} \sum_{i \in CE(e)} n_{e,i} l_e \phi_e u_e =$$
$$-\sum_{e=1}^{nEdges} l_e \phi_e u_e - l_e \phi_e u_e = 0$$

Notice that the mass conservation property is independent of temporal integration.

**4.2 Absolute Vorticity Conservation**

According to Ringler et al. (2010) formula (23), the relative vorticity is calculated according to the following diagnostic equation:

$$\xi = \frac{1}{A_v} \sum_{e \in EV(v)} t_{e,v} u_e d_e$$

Multiplying by $A_v$ and summing all of the vertices yields

$\sum_{v=1}^{nVertices} A_v \xi = \sum_{v=1}^{nVertices} \sum_{e \in EV(v)} t_{e,v} u_e d_e = \sum_{e=1}^{nEdges} \sum_{v \in VE(e)} t_{e,v} u_e d_e = \sum_{e=1}^{nEdges} u_e d_e - u_e d_e = 0$

where $e \in EV(v)$ represents the set of edges that share the vertex $v$; $v \in VE(e)$ are the two vertices on edge $e$. The indicator function $t_{e,v}$ always points to the left side of $n_{e,i}$. If $\mathbf{k} \times n_{e,i}$ is directed toward vertex $v$, then $t_{e,v} = 1$; otherwise, $t_{e,v} = -1$. $\mathbf{k}$ is the unit vector, which points in the local vertical direction. See Figure 1 for details. The total relative vorticity is shown to always be zero and independent of time.

Another method to compute the relative vorticity is to use the following prognostic equation, as described in Ringler et al. (2010) Eq. (33)

$$\frac{\partial \xi}{\partial t} + \frac{1}{A_v} \sum_{e \in EV(v)} -t_{e,v} Q_e^\perp d_e = 0$$

Multiplying the above equation by $A_v$ and summing all the vertices yields





$$\sum_{v=1}^{nVertices} A_v \frac{\partial \xi}{\partial t} = \sum_{v=1}^{nVertices} \sum_{e \in EV(v)} t_{e,v} Q_e^\perp d_e = \sum_{e=1}^{nEdges} \sum_{v \in VE(e)} t_{e,v} Q_e^\perp d_e = \sum_{e=1}^{nEdges} Q_e^\perp d_e - Q_e^\perp d_e = 0$$

Therefore, the relative vorticity is conserved during temporal integration.

Taking the partial derivative of the absolute vorticity with time yields

$$\frac{\partial \xi_a}{\partial t} = \frac{\partial \xi}{\partial t} + \frac{\partial f}{\partial t}$$

The Coriolis parameter is independent of time, $\frac{\partial f}{\partial t} = 0$; thus

$$\sum_{v=1}^{nVertices} A_v \frac{\partial \xi_a}{\partial t} = 0$$

## 5 Numerical Tests

To test the square conservation schemes using TRiSK, a new TRiSK-based shallow water dynamic core is developed, which is named TRiSK-based Multiple-Conservation dynamical cORE (TMCORE). The spatial discrete operators are the same as

those introduced by Ringler et al. (2010), the evolution variable $u_e$ is replaced by $U_e$, as we described above, and the temporal integration scheme is selected from RK or CRK, both of which are in 4$^{th}$-order precision.

We expected that the square conservation scheme will work on arbitrarily structured C-grids with a different initial field and mesh of a different resolution. In this section, we test the new scheme by using standard shallow water test cases 2, 5 and 6 (SWTC2, SWTC5, SWTC6) with two different meshes, as presented by Williamson (1992). The first mesh has 2562 Voronoi

cells (x1.2562), with an approximate resolution of 480 km, and the second mesh contains 40962 Voronoi cells (x1.40962), with an approximate resolution of 120 km. The corresponding integral time steps to x1.2562 and x1.40962 are 900 s and 360 s. Here, the midpoint scheme is selected as the method for interpolating the potential vorticity from vertices to edges for all tests.

### 5.1 Error measure methods

Global invariants error measure:

$$I(X^n) = \frac{s(x_i^n) - s(x_i^0)}{s(x_i^n)}$$

where $X_i^n$ is the variable at the $n$th time point on the $i$th cell and $X_i^0$ is the variable at the initial time. The $I$ function is the change ratio of the invariants.

The total mass error measure:

$X_i^n = h_i^n$

$Mass\ Change\ Ratio = I(h^n)$

The total energy error measure:

$X_i^n = \epsilon_i^n$

$Energy\ Change\ Ratio = I(\epsilon^n)$





Measuring the fluid thickness error by $L_2$ and $L_\infty$ error norms is expressed as

$$L_2 = \frac{\left\{S\left[\left(f_m(i) - f_R(i)\right)^2\right]\right\}^{\frac{1}{2}}}{\left[S(f_R(i)^2)\right]^{\frac{1}{2}}}$$

$$L_\infty = \frac{\max|f_m(i) - f_R(i)|}{\max|f_R(i)|}$$

where $i$ denotes the index of each cell; $f_m(i)$ and $f_R(i)$, respectively, are the model solution and reference solution at the ith

cell on the mesh; and the $S$ function is the area-weighted accumulation of an arbitrary variable $X$.

$$S(X) = \frac{\sum_{i=1}^{N} X(i)A(i)}{\sum_{i=1}^{N} A(i)}$$

where $A(i)$ is the area of the $i$th cell.

The reference solution should be an analytical solution or, when an analytical solution is not available, a high-resolution solution from the model with sufficient accuracy.

In the following context, CRK4 represents the CRK with 4th-order precision and RK4 represents the original Runge-Kutta scheme with 4th-order precision.

The differences of the error norms between CRK4 and RK4 schemes by using the different ratios of $L_2$ (L2DR) and $L_\infty$ (LInfDR) is expressed as:

$$L2DR = \frac{L_{2_{CRK4}} - L_{2_{RK4}}}{L_{2_{RK4}}}$$

$$LInfDR = \frac{L_{\infty_{CRK4}} - L_{\infty_{RK4}}}{L_{\infty_{RK4}}}$$

where $L_{2_{CRK4}}$ and $L_{2_{RK4}}$ are the $L_2$ error norms of CRK4 and RK4, respectively, which is similar for $L_{\infty_{CRK4}}$ and $L_{\infty_{RK4}}$. CRK4 has better performance than RK4 when the different ratios are negative; otherwise, CRK4 has worse performance than RK4.

## 5.2 Global Steady State Nonlinear Zonal Geostrophic Flow (SWTC2)

For the Global Steady State Nonlinear Zonal Geostrophic Flow test case, the initial velocity field has the following form

$$u = u_0 \cos\theta$$

$$v = 0$$

The geopotential height field is

$$gh = gh_0 - \left(a\Omega u_0 + \frac{u_0^2}{2}\right)\sin^2\theta$$

Here, we set $\Omega = 7.292 \times 10^{-5}\ s^{-1}, g = 9.80616\ m/s^2, a = 6.37122 \times 10^6\ m, gh_0 = 2.94 \times 10^4\ m^2/s^2$ and $u_0 = 2\pi a/(12\ days)$, where $\theta$ denotes latitude. In this test case, the exact solution is the initial state, and any difference between the numerical solution and the initial state is the simulation error.



In SWTC2, the true solution of $\frac{\partial u}{\partial t}, \frac{\partial v}{\partial t}, \frac{\partial \phi}{\partial t}$ is always zero; therefore, this test case can only represent the precision of spatial discrete operators but not the precision of temporal integration. This simulation is integrated for 10 years, but the shape of geostrophic flow breaks after 7 years. Therefore, we choose the simulation results from the 1$^{st}$ to the 7$^{th}$ year to compare the error norms of CRK4 and RK4.

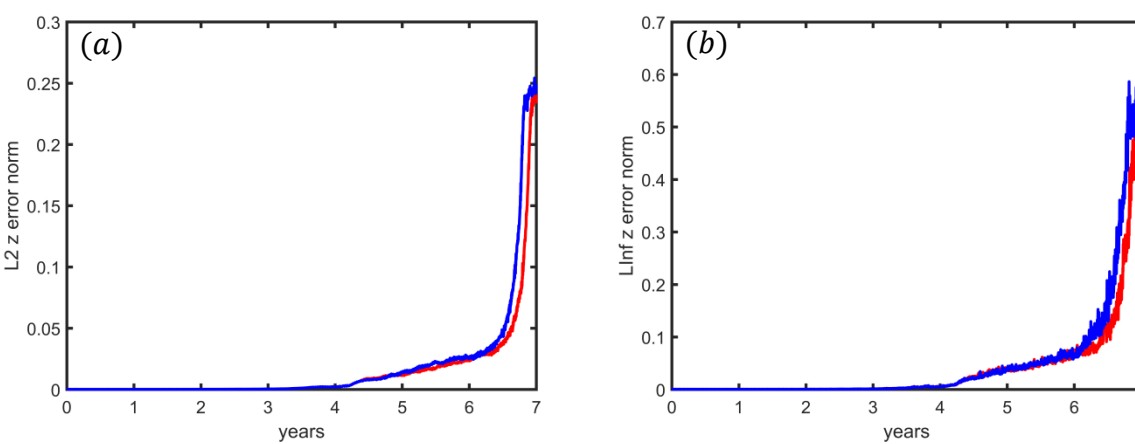

**Figure 2.** Geopotential height error norms of SWTC2. (a) $\mathbf{L_2}$ error norm; (b) $\mathbf{L_\infty}$ error norm. The results of RK4 and CRK4 are represented by blue and red lines, respectively. The model mesh is x1.2562.

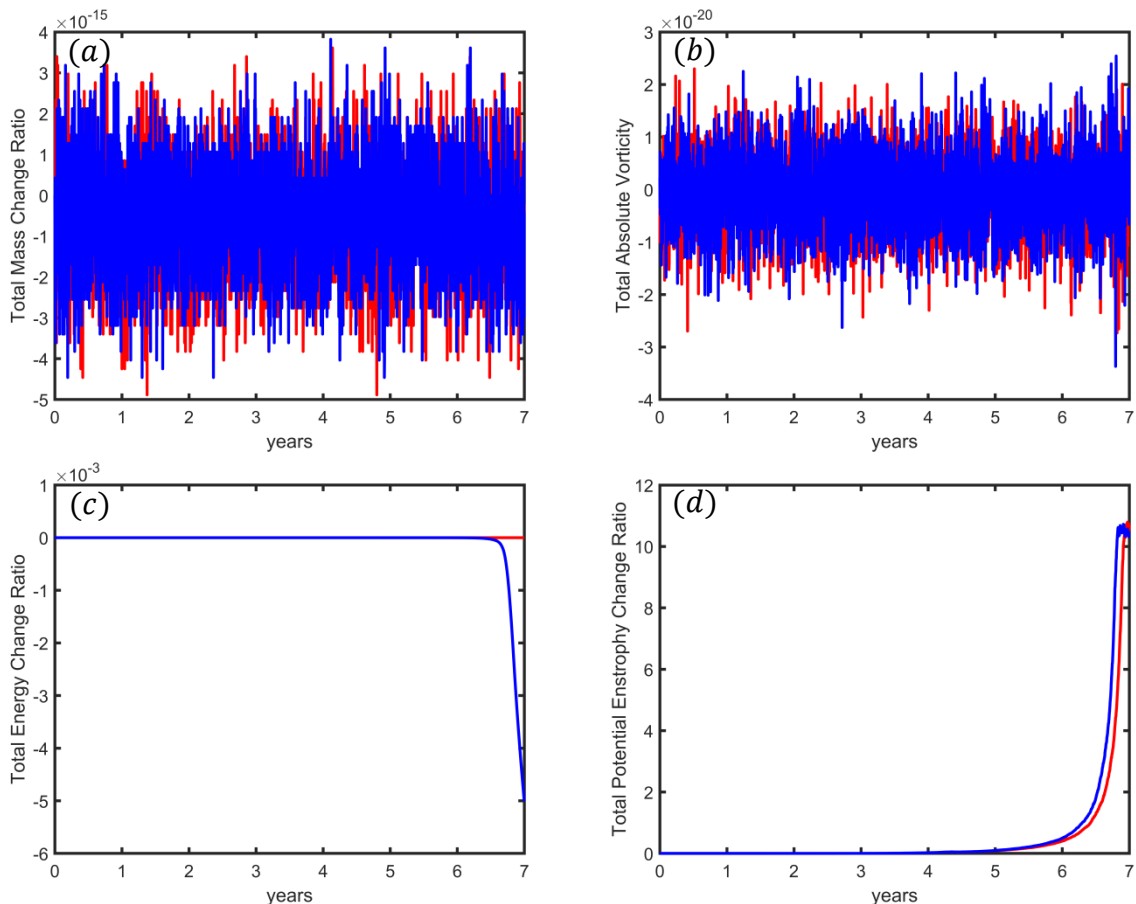

**Figure 3.** The variation of integral invariants as a function of time of SWTC2. (a) Total mass change ratio; (b) total absolute vorticity; (c) total energy change ratio; (d) total potential enstrophy change ratio. The results of RK4 are represented by blue lines; the results of CRK4 are represented by red lines. The model mesh is x1.2562.

Figure 2 measures the $L_2$ and $L_\infty$ error norms of geopotential height. In the first 4 years, the CRK4 and RK4 exhibit similar results, but in the last 3 years, the shape of geopotential flow tends to break. The error norms increase sharply after 6 years, and the differences between CRK4 and RK4 become more evident. Both the $L_2$ and $L_\infty$ error norms of CRK4 are evidently smaller than RK4, and the collapse of geopotential flow is delayed approximately 1 month when using CRK4.

Figure 3 presents the variation of invariants as a function of time. The oscillations of total mass and total absolute vorticity are strictly conserved. The total energy of RK4 decreased approximately 0.5% in the final year, but CRK4 maintains strict energy conservation. Although the geopotential flow has been broken, CRK4 prevents an increasing rate of total potential enstrophy.

### 5.3 Zonal Flow Over an Isolated Mountain (SWTC5)

SWTC5 is the $5^{th}$ test case described by Williamson 1992; the wind and height fields are similar to SWTC2, but $h_0 = 5960\ m, u_0 = 20\ m/s$ and mountain height is determined according to the following equation:





$$h_s = h_{s0}\left(1 - \frac{r}{R}\right)$$

where $h_{s0} = 2000\ m$; $R = \frac{\pi}{9}$; $r = \sqrt{min[R^2, (\lambda - \lambda_c)^2 + (\theta - \theta_c)^2]}$; and $\lambda_c$ and $\theta_c$ are the center longitude and latitude of the mountain, respectively. Here, we set $\lambda_c = \frac{3\pi}{2}$ and $\theta_c = \frac{\pi}{6}$. As the analytical solution is not available, the reference solution is provided by a T511 idealized global spectral atmospheric model from GFDL, where $8 \times 10^{12}\ m^4/s$ is selected as the

coefficient for the $\nabla^4$ dissipation, and the test case is integrated for 50 days.

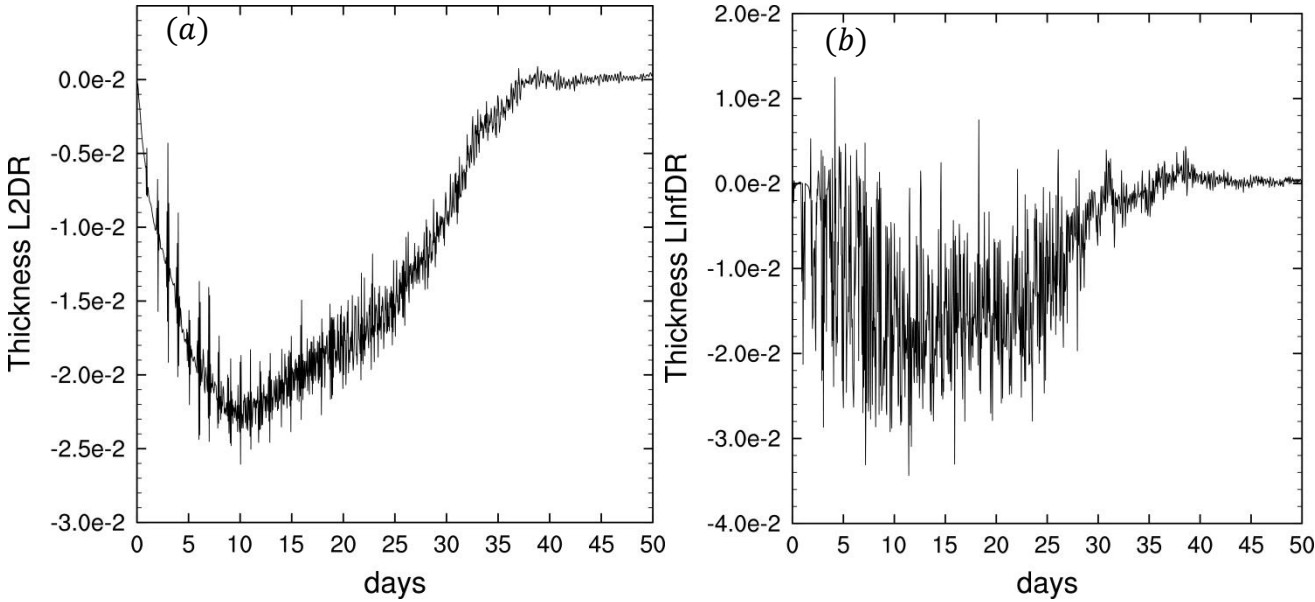

**Figure 4.** Fluid thickness error norms of different SWTC5 ratios. (a) $\mathbf{L_2}$ error norm difference ratio; (b) $\mathbf{L_\infty}$ error norm difference ratio. The model mesh is x1.40962.



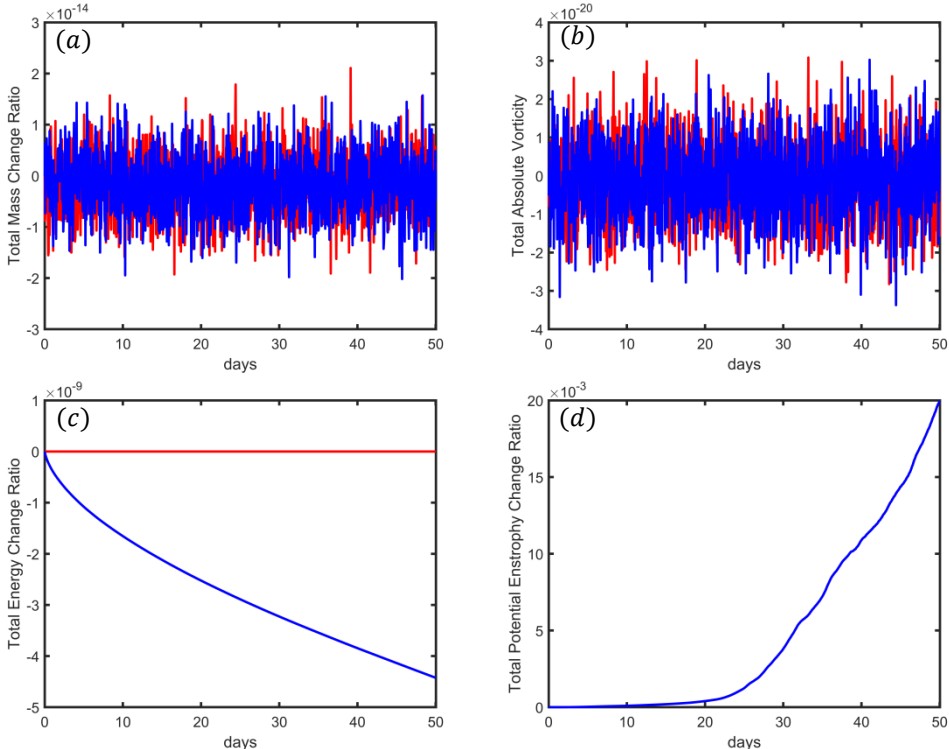

**Figure 5.** Integral invariants of SWTC5. (a) Total mass change ratio; (b) total absolute vorticity; (c) total energy change ratio; (d) total potential enstrophy change ratio. The results of RK4 and CRK4 are represented by blue and red lines, respectively. The model mesh is x1.40962.

Figure 4 presents the different ratios of error norms. In the first 35 days, the $L_2$ and $L_\infty$ error norms of CRK4 are considerably smaller than those of RK4. Compared with RK4, the $L_2$ error norm of CRK4 decreases by approximately 2.5% at the minimum point of L2DR, and the $L_\infty$ error norm also decreases by approximately 3% at the minimum point of LInfDR. The error norms increase very quickly after 35 days; therefore, the differences between error norm ratios for CRK4 and RK4 tend to be similar, along with time.

Figure 5 presents the variation of the invariants as a function of time. The total mass and total absolute vorticity are completely conserved for both CRK4 and RK4. CRK4 is able to maintain strict energy conservation from the beginning to the end, but the total energy of RK4 is dissipative. The CSCS exhibits no influence on the total potential enstrophy.

**5.4 Rossby-Haurwitz Wave (SWTC6)**

The classical 4 zonal wavenumber Rossby-Haurwitz wave was selected as the third test case. The initial condition follows Williamson (1992). The initial state is the analytical solution of the nonlinear barotropic vorticity equation on the sphere but not the analytical solution of the shallow water equations. The reference field is provided by a T511 idealized global spectral



atmospheric model from GFDL. To limit the noise of the spectral model, we use $5 \times 10^{12}$ m$^4$/s as the coefficient for the $\nabla^4$ dissipation. As presented by Williamson, 1992, the phase speed of the Rossby-Haurwitz wave is calculated as follows:

$$c = \frac{R(R+3)\omega - 2\Omega}{(R+1)(R+2)}$$

where $R = 4$ is the zonal wavenumber of the Rossby-Haurwitz wave; $\omega = 7.848 \times 10^{-6} s^{-1}$; and $\Omega = 7.292 \times 10^{-5} \ s^{-1}$ is the rotation rate of the earth; therefore, the 4 zonal wavenumber period of the Rossby-Haurwitz wave is approximately 29.52 days. We integrate the test case over one period (33 days).

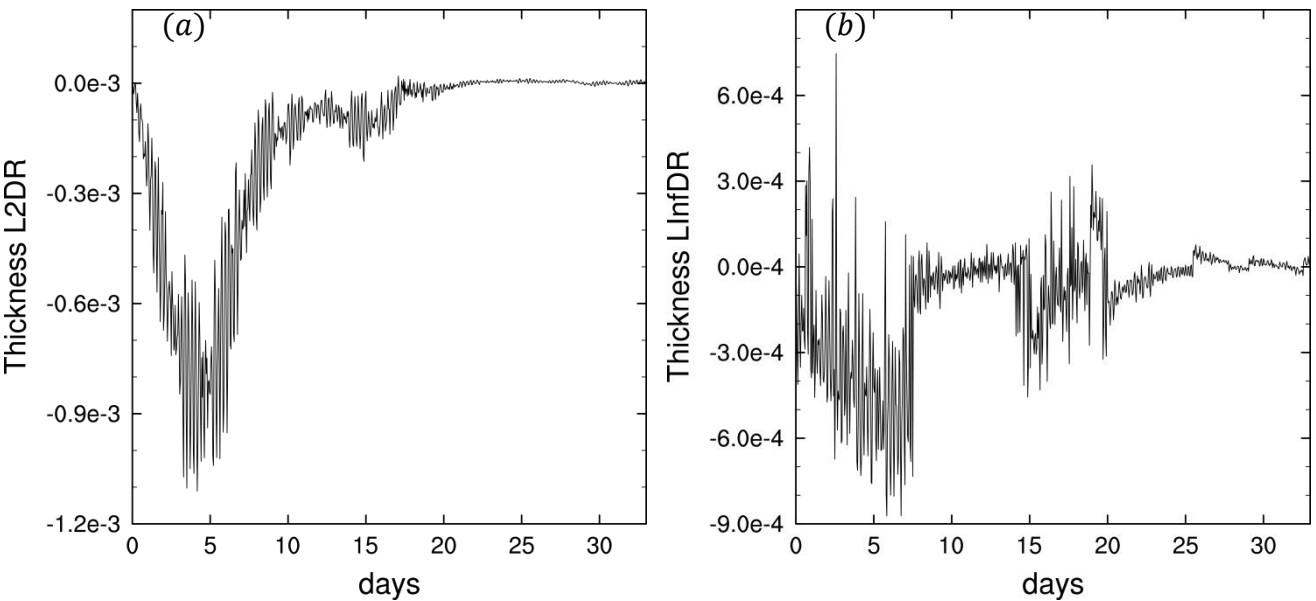

**Figure 6.** The fluid thickness error norms of different SWTC6 ratios. (a) $L_2$ error norm difference ratio; (b) $L_\infty$ error norm difference ratio. The model mesh is x1.40962.





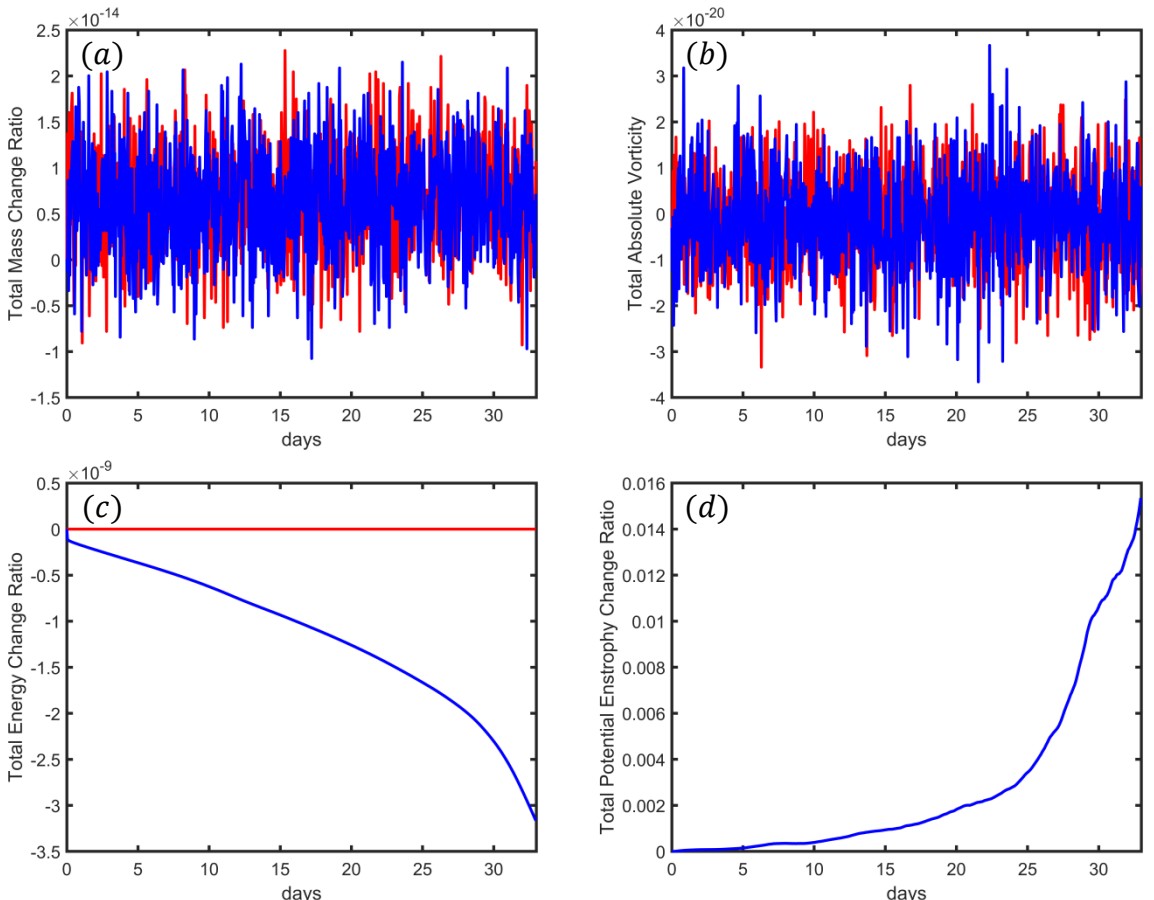

**Figure 7.** Integral invariants of SWTC6. (a) Total mass change ratio; (b) total absolute vorticity; (c) total energy change ratio; (d) total potential enstrophy change ratio. The results of RK4 and CRK4 are represented by blue and red lines, respectively. The model mesh is x1.40962.

In both simulations of CRK4 and RK4, the Rossby-Haurwitz wave begins to distort at the 25[th] day and then collapse a few days later.

Figure 6 presents the error norm difference ratios. CRK4 has a smaller $L_2$ error norm than RK4 in the first 20 days. With growth of the $L_2$ error norm, the difference between CRK4 and RK4 trends toward zero. At the 4[th] day, the $L_2$ error norm of CRK4 is more than 0.11% less than that of RK4. CRK4 also has a smaller $L_\infty$ error norm a majority of the time. At the 6[th] day, the $L_\infty$ error norm of CRK4 is more than 0.08% less than that of RK4.

Figure 7 presents the variation of invariants as a function of time. The total mass and total absolute vorticity are strictly conserved for both CRK4 and RK4. As expected, CRK4 maintains strict energy conservation, and RK4 cannot conserve the total energy during integration. With the Rossby-Haurwitz wave collapse, the total energy of RK4 rapidly dissipates after 25 days. There is no influence of CRK4 to potential enstrophy in this case.



## 6 Summary

In this paper, we extend the CSCS to arbitrarily structured C-grids with shallow water equations, and we estimate the performance of the CSCS by using standard shallow water test cases.

There are two prerequisites for constructing CSCS, the anti-symmetrical spatial discrete operator and the square conservative temporal integration scheme. It is difficult to directly construct an anti-symmetrical spatial discrete operator on quasi-uniform grids; therefore, we take advantage of the instantaneous energy conservation property of the spatial discrete operators, as described by Ringler et al. (2010), to obtain the anti-symmetrical operator. After the IAP transformation, the units of evolution variables are unified, and the evolution variable $u_e$ is replaced with $U_e = \sqrt{\phi_e}u_e$. According to the derivative rule (19), the temporal trend of $U_e$ is expressed as a combination of the temporal trends of $u_e$ and $\phi_i$, and we demonstrate that the spatial discrete operator of $U_e$ is an anti-symmetrical operator. Then, we integrate the model with the square conservative temporal integration scheme CRK4, and the complete square conservation property is achieved.

An important finding is the equivalency between the energy conservative operator and anti-symmetrical operator for both the continuous system and discrete system. In most of previous study, anti-symmetrical operators were constructed on uniform grids, especially longitude–latitude grids, and the equation's advection term was in the advection-flux form. We extend the square conservation theory to a more general situation. The anti-symmetrical spatial discrete operator is constructed on quasi-uniform grids, and the equation is in the vector-invariant form.

The CSCS is able to maintain three integral invariants, including total mass, total absolute vorticity and total energy, in all the test cases, and the error norms decrease in varying degrees. The square conservation scheme improves the stability in SWTC2, and the error norms of CRK4 are evidently less than RK4 after 4 years of simulation. For RK4, the total energy dissipates very quickly after the geostrophic flow collapse, but CRK4 maintains complete energy conservation for the entire period, and the increasing rate of the total potential enstrophy is also limited by the square conservation scheme. In both SWTC5 and SWTC6, CRK4 not only maintains strict conservation for three integral invariants but also leads to less error norms than RK4.

## Appendix A

In this appendix, we attempt to demonstrate that the spatial discrete operator $L$ is energy conservative. Our objective is to prove that the following equation is able to be satisfied by $L$,

$$\left(U, \frac{\partial U}{\partial t}\right)_e = \left(\phi, \frac{\partial \phi}{\partial t}\right)_i$$

Consider the inner product on edge

$$\left(U, \frac{\partial U}{\partial t}\right)_e = \sum_{e=1}^{nEdges} U_e \frac{\partial U_e}{\partial t} A_e$$

Substituting (19) into above formula

$$\left(U, \frac{\partial U}{\partial t}\right)_e = \sum_{e=1}^{nEdges} U_e \left(C_e \frac{\partial u_e}{\partial t} + \frac{u_e}{2h_e} \frac{\partial \phi_e}{\partial t}\right) A_e$$





Where $C_e = \sqrt{\phi_e}$ is the phase speed of external-gravity wave on edges.

According to Eq. (52) in Ringler et al. (2010),

$$\left(U, \frac{\partial U}{\partial t}\right)_e = \sum_{e=1}^{nEdges} \left(\phi_e u_e \frac{\partial u_e}{\partial t} + \frac{u_e^2}{4} \sum_{i \in CE(e)} \frac{\partial \phi_i}{\partial t}\right) A_e$$

According to Eq. (63) and (A.8) in Ringler et al. (2010),

$$\left(U, \frac{\partial U}{\partial t}\right)_e = \sum_{e=1}^{nEdges} \phi_e u_e \frac{\partial u_e}{\partial t} A_e + \sum_{i=1}^{nCells} K_i \frac{\partial \phi_i}{\partial t} A_i$$

Substituting (4) into above formula

$$\left(U, \frac{\partial U}{\partial t}\right)_e = \sum_{e=1}^{nEdges} \phi_e u_e A_e \left(Q_e^\perp + \frac{1}{d_e} \sum_{i \in CE(e)} n_{e,i} E_i\right) + \sum_{i=1}^{nCells} K_i \frac{\partial \phi_i}{\partial t} A_i$$

According to section 3.7.2 in Ringler et al. (2010),

$$\sum_{e=1}^{nEdges} \phi_e u_e A_e Q_e^\perp = 0$$

Since $A_e = l_e d_e$

$$\left(U, \frac{\partial U}{\partial t}\right)_e = \sum_{e=1}^{nEdges} \phi_e u_e l_e \sum_{i \in CE(e)} n_{e,i} E_i + \sum_{i=1}^{nCells} K_i \frac{\partial \phi_i}{\partial t} A_i$$

where $E_i = K_i + \phi_i$.

According to (A.4) in Ringler et al. (2010),

$$\left(U, \frac{\partial U}{\partial t}\right)_e = \sum_{i=1}^{nCells} E_i \sum_{e \in EC(i)} n_{e,i} \phi_e u_e l_e + \sum_{i=1}^{nCells} K_i \frac{\partial \phi_i}{\partial t} A_i$$

According to (5),

$$-A_i \frac{\partial \phi_i}{\partial t} = \sum_{e \in EC(i)} n_{e,i} l_e \phi_e u_e$$

Therefore,

$$\left(U, \frac{\partial U}{\partial t}\right)_e = -\sum_{i=1}^{nCells} E_i \frac{\partial \phi_i}{\partial t} A_i + \sum_{i=1}^{nCells} K_i \frac{\partial \phi_i}{\partial t} A_i$$

Consider the inner product on cell

$$\left(\phi, \frac{\partial \phi}{\partial t}\right)_i = \sum_{i=1}^{nCells} \phi_i \frac{\partial \phi_i}{\partial t} A_i$$

Thus,

$$\left(U, \frac{\partial U}{\partial t}\right)_e + \left(\phi, \frac{\partial \phi}{\partial t}\right)_i = -\sum_{i=1}^{nCells} E_i \frac{\partial \phi_i}{\partial t} A_i + \sum_{i=1}^{nCells} K_i \frac{\partial \phi_i}{\partial t} A_i + \sum_{i=1}^{nCells} \phi_i \frac{\partial \phi_i}{\partial t} A_i = 0$$

*Code availability.* Idealized Global Spectral Atmospheric Models (GFDL): https://www.gfdl.noaa.gov/idealized-spectral-
models-quickstart/ (last access: 3 May 2019). TMCORE is available at https://github.com/TMCORE-Project/TMCORE (last access: 3 May 2019). The digital object identifier for Idealized Global Spectral Atmospheric Models (GFDL) with standard shallow water test cases is http://doi.org/10.5281/zenodo.3249878. The digital object identifier for TMCORE v1.0 is http://doi.org/10.5281/zenodo.3241647.





*Author contributions.* Lilong Zhou developed the theory, wrote the initial version of TMCORE, implemented the experiments and wrote the manuscript. Jiming Feng and Lijuan Hua revised the context structure of the manuscript and gave some useful technical advices.

*Competing interests.* The authors declare that they have no conflict of interest.

*Acknowledgements.* This work was supported by the National Key R&D Program of China (2016YFA0600403) and the General Project of the National Natural Science Foundation of China (Grant 41875134).

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
