# Peer review of "Extending Square Conservation to Arbitrarily Structured C-grids with Shallow Water Equations"

_Geoscientific Model Development, 2019_

## Referee Comment (RC1) · Anonymous Referee #1 · 19 Aug 2019

General comments

The manuscript considers the numerical solution of shallow water equations on quasi-uniform spherical meshes. It explains how an energy-conserving spatial discretization can be combined, using a suitable change of prognostic variables, with a square-conserving temporal discretrization to obtain a scheme that is exactly energy conserving.

There are some problems with the presentation in the manuscript (details below), which it should be possible to fix. However, the key ideas needed to obtain exact energy conservation on arbitrary meshes have been around for a while; this paper merely brings them togther. Also, temporal truncation errors tend to be much smaller than spatial truncation errors in atmospheric models, so only a small improvement (if

any) is obtained by replacing an energy-conserving spatial discretization by an energy-conserving space-time discretization, (as the results in this paper confirm). Thus, I think the manuscript lacks the originality and significance needed to justify publication in GMD.

—

Specific comments

Sections 3.1, 3.2. The notation $\mathcal{L}F$, $\mathcal{M}u$, $\mathcal{N}\phi$ suggests that $\mathcal{L}$, $\mathcal{M}$, $\mathcal{N}$ are linear functions of $F$, $u$, and $\phi$ respectively. In fact they are all nonlinear functions. Moreover, $\mathcal{M}$ and $\mathcal{N}$ are actually functions of both $u$ and $\phi$. These two sections are over-elaborate and presented in a very confusing way. In several places it is not obvious what is assumed and what is claimed to be proved. All that is really needed is the fact that the energy at a point can be written as a squared quantity by making a certain change of variables.

P1 line 27, also P2 lines 3-5. The opening sentence is too categorical. For a quantity like energy or potential enstrophy, in a numerical model the total is made up of resolved and unresolved contributions. Therefore it is not obvious that conserving the resolved contribution is necessary for a good solution; indeed it may be necessary to dissipate the resolved contribution (e.g. to prevent 'spectral blocking'). One can argue for a conservative numerical method by saying that we want to parameterize any dissipation, not leave it to numerical errors, but the opposite argument can also be made. Such ideas are extensively discussed in the literature.

P1 line 29, Figs. 3b, 5b, 7b, P19 line 17. On a spherical domain the vanishing of the global integral of vorticity is a purely kinematic identity. Provided the vorticity and its integral are calculated in a self-consistent way, the same result must hold in the discrete case (e.g. P10 line 20). Thus conservation of the global integral of vorticity is a test only of the fact that the vorticity is calculated consistently; it says nothing about the properties of the numerical methods used to solve the equations.

P1 line 29. '...five basic physical conserved properties'. Actually potential enstrophy is just one member of an infinite family (so-called Casimirs).

P3 line 23. This form of the equations is usually called 'vector-invariant' (as on P19 line 16).

P4. Equation (3) is inconsistent with the definition of the 2-norm on line 5. This paragraph seems to be mixing up point values and global integrals.

Figure 1. Note that the grid used need not be uniform and regular (as suggested by the figure).

P5. Note that the sign convection for $u_e$ is related to the direction of the unit normal - this is crucial to get everything to work out. Also crucial for energy conservation is that $Q_e^{\perp}$ is constructed to satisfy the equation on P20 line 9.

P5 line 22. '...new type of Runge-Kutta'. Not so new (1996).

P6 line 11. It would be helpful to give a reference for 'IAP transformation'.

P11 lines 22-23. The text does not make sense - it seems to be mixing up point values and global integrals.

Section 5.2. The fact that the solid body rotation flow eventually breaks down, despite the conservation properties of the scheme, is intriguing. Could this be a manifestation of the 'Hollingsworth instability' (as discussed, for example, in Skamarock et al 2012)?

P19 line 26. Sign error? Line 30. What is $h_e$ ?

References are not in alphabetical order.

---

## Referee Comment (RC2) · Anonymous Referee #2 · 21 Aug 2019

General comments

This manuscript proposed a square conservation scheme for improving the TRiSK shallow water dynamic core on quasi-uniform grids. This scheme includes an anti-symmetrical spatial discrete operator and a temporal integral scheme with exact quadratic conservation in matematics. The improved dynamic core with the new scheme conserves three physical integrals including the total energy, total mass and total vorticity, and reduces the simulation errors in numerical tests comparing with the original dynamic core.

The method to implement the anti-symmetry of the spatial discrete operator of the improved dynamic core is unique and economical, which cleverly uses a simple combination of the original spatial discrete operator based on the IAP transformation. The

temporal integration applies the improved Runger-Kutta scheme with exact quadratic conservation proposed by Wang et al (1996), and thus keeps the energy conservation law in physics that is equal to the square conservation in mathematics. As I know, the manuscript presents one of the earliest works of implementing the total energy conservation of a shallow water dynamic core on quasi-uniform grids in the way of quadratic conservation in mathematics, which is quite different from the available and similar works that use the energy equation to replace the continuity equation.

However, the presentation of the manuscript needs improvements because of some incorrect mathematic equalities and improper expressions in the text.

Specific comments

1) The necessity to conserve the resolved energy in numerical solutions to an energy conservation system is actually the same as that to conserve the resolved mass. To highlight the significance of constructing an energy conservation scheme for the TRiSK dynamic core, a clear explanation on the necessity should be provided in Section 1. 2) Line 19/Page 2: CRK is improperly used as the abbreviation of "a new class of Runger-Kutta scheme", because the word "class" does not describe the main characteristics of this scheme. NRK is better. 3) I wonder why the title of Section 2 is exactly the same as that of Section 1 (Lin 22/Page 3). 4) The equality (3) (Line 4/Page 4) is not true, which missed the integration sign after the second equal mark. 5) The semi-discrete form of the shallow water equation set [Equations (4)-(5) on Lines 4-5/Page 5] should no longer be a partial differential equation set, but an ordinary differential equation set. 6) Line 6/Page 5: u and v are not the variables of Eqs.(4)-(5). 7) Line 20/Page 7: The equality is not true, because a negative sign is missed before(âĎŠðİŚ■, ðİŚ■).

Minor comments

8) Line 10/Page 1: "The square conservation theory is widely used on latitude-longitude grids" –> "The square conservation law is maintained in the dynamic cores on latitude-longitude grids". 9) Line 4/Page 2: "which is" –> "which are". 10) Line 26/Page 2: "polar

problem" –> "polar instability" or "polar singularity".

---

## Author Comment (AC1) · 2 Sep 2019

Thank you very much for reading our manuscript meticulously, we put the replies in the supplement pdf, and there are references in the zip file.

Please also note the supplement to this comment:
https://www.geosci-model-dev-discuss.net/gmd-2019-122/gmd-2019-122-AC1-supplement.zip
* * *

---

## Author Comment (AC2) · 2 Sep 2019

We would like to thank you for the positive comments and constructive advices, which help us to make the manuscript more clearly and more persuasive. The responses for the comments are in the supplement zip file.

Please also note the supplement to this comment: https://www.geosci-model-dev-discuss.net/gmd-2019-122/gmd-2019-122-AC2-supplement.zip
* * *

---

## Author Comment (AC4) · 2 Sep 2019

[revised manuscript text omitted]
( \left\| \boldsymbol{\phi} \boldsymbol{K} \right\| + \left\| \frac{1}{2} \boldsymbol{\phi}^{2} \right\| + \left\| \boldsymbol{\phi} \boldsymbol{\phi}_{s} \right\| \right) \\ &= \frac{de}{dt} = 0 \end{aligned}$$

Accordingly, square conservation is equivalent to energy conservation in a continuous system.

**3.2 Constructing the anti-symmetrical spatial discrete operator**

In this subsection, we construct the anti-symmetrical spatial discrete operator by a specific combination of original operators in TRiSK. Firstly, we need to demonstrate the equivalent relationship between square conservation and energy conservation in continuous system, then prove that this relationship is also apply to discrete system.

5 Assuming a continuous-in-time system, the evolution equation of U is able to be expressed as

$$\frac{\partial U}{\partial t} = \sqrt{\phi} \frac{\partial u}{\partial t} + \frac{u}{2\sqrt{\phi}} \frac{\partial \phi}{\partial t},\tag{13}$$

This formula is key to connecting square conservation and energy conservation; it is difficult to directly construct the antisymmetrical operator on quasi-uniform grids.

Theorem. The operators functions  $\mathcal{M} = \mathcal{M}(\phi, u)$  and  $\mathcal{N} = \mathcal{N}(\phi, u)$  satisfy

10
$$\begin{cases} \frac{\partial u}{\partial t} + \mathcal{M}_{tt} = 0\\ \frac{\partial \phi}{\partial t} + \mathcal{N}_{tt} = 0 \end{cases}$$
 (14)

and

$$(\mathcal{M}_{\boldsymbol{u}}, \boldsymbol{\phi}\boldsymbol{u}) + (\mathcal{N}_{\boldsymbol{\phi}}, \boldsymbol{E}) = 0$$

After IAP transformation (9), the evolution equation of U may be expressed as (13), and (14) may be rewritten as (10). If the operator  $\mathcal{L}$  satisfies (10), then  $\mathcal{L}$  is an anti-symmetrical operator.

15 Proof.

$$\begin{aligned} \frac{\partial U}{\partial t} &= \sqrt{\phi} \frac{\partial u}{\partial t} + \frac{u}{2\sqrt{\phi}} \frac{\partial \phi}{\partial t} = -\sqrt{\phi} \mathcal{M}_{\underline{t}\underline{t}} - \frac{u}{2\sqrt{\phi}} \mathcal{N}\phi \\ (\mathcal{L}(F), F) &= -\left(\frac{\partial U}{\partial t}, U\right) + -\left(\frac{\partial \phi}{\partial t}, \phi\right) \\ &= \oint_{\Omega} - U \frac{\partial U}{\partial t} - \phi \frac{\partial \phi}{\partial t} \, ds \\ &= \oint_{\Omega} - U \left(-\sqrt{\phi} \mathcal{M}_{\underline{t}\underline{t}} - \frac{u}{2\sqrt{\phi}} \mathcal{N}\phi\right) - \phi \mathcal{N}\phi \, ds \\ 20 &= \oint_{\Omega} -\phi u \cdot \mathcal{M}_{\underline{t}\underline{t}} + -\frac{|u|^2}{2} \mathcal{N}\phi + -\phi \mathcal{N}\phi \, ds \\ &= -(\mathcal{M}_{\underline{t}\underline{t}}, \phi u) + -(\mathcal{N}\phi, E) \\ &= 0 \end{aligned}$$

This theorem is proved in a continuous system, but the model is built in a discrete system; therefore, it is necessary to discuss the situation in a discrete system.

25 Following Ringler et al. (2010), we set the discrete operators functions  $M = M(\phi, u)$  and  $N = N(\phi, u)$  as:

 $M_{\boldsymbol{u}}^{\boldsymbol{u}} = [\nabla E]_{e} - Q_{e}^{\perp}$  $N_{\boldsymbol{\phi}}^{\boldsymbol{\phi}} = [\nabla \cdot (\boldsymbol{\phi} u)]_{i}$

And the semi-discrete shallow water equation set becomes

$$\frac{\partial u}{\partial t} + M_{tt} = 0, \qquad (15)$$

$$\frac{\partial \phi}{\partial t} + N\phi = 0, \qquad (16)$$

Because the semi-spatial discrete operator of TRiSK has an instantaneous energy conservation property, it is easy to prove  $(M_{u}, \phi_u) + (N_{\phi}, E) = 0$ . (Details in Ringler et al. (2010), section 3.7.2)

[revised manuscript text omitted]

In the following context, CRK4-NRK4 represents the CRKNRK with 4th-order precision and RK4 represents the original Runge-Kutta scheme with 4th-order precision.

The differences of the error norms between CRK4NRK4 and RK4 schemes by using the different ratios of  $L_2$  (L2DR) and  $L_{\infty}$  (LInfDR) is expressed as:

$$L2DR = \frac{L_{2CRK4} - L_{2RK4}}{L_{2RK4}}$$
$$LInfDR = \frac{L_{\infty CRK4} - L_{\infty RK4}}{L_{\infty RK4}}$$

15 where  $L_{2_{CRK4}}$  and  $L_{2_{RK4}}$  are the  $L_2$  error norms of CRK4NRK4 and RK4, respectively, which is similar for  $L_{\infty_{CRK4}}$  and  $L_{\infty_{RK4}}$ . <del>CRK4NRK4</del> has better performance than RK4 when the different ratios are negative; otherwise, CRK4NRK4 has worse performance than RK4.

**5.2 Global Steady State Nonlinear Zonal Geostrophic Flow (SWTC2)**

For the Global Steady State Nonlinear Zonal Geostrophic Flow test case, the initial velocity field has the following form

 $20 \quad u = u_0 \cos \theta$

v = 0

The geopotential height field is

 $gh = gh_0 - \left(a\Omega u_0 + \frac{u_0^2}{2}\right)\sin^2\theta$

Here, we set  $\Omega = 7.292 \times 10^{-5} s^{-1}$ ,  $g = 9.80616 m/s^2$ ,  $a = 6.37122 \times 10^6 m$ ,  $gh_0 = 2.94 \times 10^4 m^2/s^2$  and  $u_0 = 2\pi a/(12 \ days)$ , where h is fluid thickness,  $\theta$  denotes latitude. In this test case, the exact solution is the initial state, and any difference between the numerical solution and the initial state is the simulation error.

In SWTC2, the true solution of  $\frac{\partial u}{\partial t}$ ,  $\frac{\partial v}{\partial t}$ ,  $\frac{\partial \phi}{\partial t}$  is always zero; therefore, this test case can only represent the precision of spatial discrete operators but not the precision of temporal integration. This simulation is integrated for 10 years, but the shape of

geostrophic flow breaks after 7 years. Therefore, we choose the simulation results from the  $1^{st}$  to the 7th year to compare the error norms of CRK4NRK4 and RK4.

---

## Author Response (AR1)

**Author's Response**

First of all, we'd like to thank the referees and editors, the comments and advices they mentioned help us a lot to improve the manuscript. We modify the manuscript with more general description and some of our new insight.

**Replys to Anonymous Referee #1**

5 Thank you very much for reading our manuscript meticulously, those problems you found and the advices you mentioned, help us a lot to improve the description and strictness to the manuscript. There are some opinions we'd like to discuss and share with you.

The main theme of this manuscript is finding out a method to extend the square conservation scheme, from regular latitudelongitude grid to an arbitrarily structured C-grids dynamic core, TRiSK, meanwhile, the intrinsic property of TRiSK (including

10 accuracy of operators and conservation properties) should not be broken down.

There are some problems with the presentation in the manuscript (details below), which it should be possible to fix. However, the key ideas needed to obtain exact energy conservation on arbitrary meshes have been around for a while; this paper merely brings them together. Also, temporal truncation errors tend to be much smaller than spatial truncation errors in atmospheric

15 models, so only a small improvement (if any) is obtained by replacing an energy-conserving spatial discretization by an energyconserving space-time discretization, (as the results in this paper confirm). Thus, I think the manuscript lacks the originality and significance needed to justify publication in GMD.

Reply:

The most of so-called energy conservation schemes on arbitrary meshes merely conserve total energy within time truncation

- 20 error, i.e. (Ringler et al,2010), (Thi-Thao-Phuong Hoang,2019), the energy is still slightly dissipative during temporal integration. As Eq.(24) in the manuscript, the energy is strictly conserved only if  $2\tau_n(\varphi^n, F^n) + \tau_n^2 ||\varphi^n||^2 = 0$ , unfortunately, most of temporal integration schemes do not satisfy this condition. Conserving energy within time truncation error is not equivalent to strict/exact energy conservation, the former allows slightly energy dissipative/anti-dissipative during temporal integration, the later conserving total energy in round-off error. In the manuscript, Figure 3c, Figure 4c and Figure 5c (the
- 25 same as the following figures, from left to right) are showing the differences between "Conserving energy within truncation error" (blue line) and "Strictly conserving energy" (red line).

Indeed, there are some methods to exactly conserve energy on arbitrary meshes, i.e. taking energy as an evolution variable, based on conservation law, the energy flux is balanced between each cell, therefore energy is conserved anywhere and anytime (Satoh, 2004, Section 1.2.3). But these methods are not quadratic conservation in mathematics. In the shallow water system,

- 5 one can obtain the exact energy conservation by replacing the continuous equation by energy equation, but this method sacrifices mass conservation; or in another way, replacing the momentum equation by energy equation, but the flow direction will not able to be determined, and sometimes worse situation appears, since the lack of constrain from momentum equation, potential energy could be greater than total energy, which result in the wind speed becomes imaginary number.
- By implementing the square conservation scheme, neither momentum equation nor continuous equation needs to be replaced
  by energy equation, the total energy is strictly conserved, rather than conserved within time truncation error, meanwhile, there are not influences to the other conservative properties, such as mass and absolute vorticity.
  About the originality. The prerequisite of implementing square conservation scheme is that the spatial discrete operator must be anti-symmetrical, but it is hard to construct an anti-symmetrical operator on quasi-uniform grids directly, therefore we try to find another way to obtain the anti-symmetrical operator. Energy conservation is one of the intrinsic properties of TRiSK
- 15 shallow water dynamic core, and as we mentioned in Section 3.2, Eq.(13) is the key to connecting square conservation and energy conservation, by using this simple combination of original TRiSK spatial discrete operators, the anti-symmetrical operator is built.

Note that, for constructing the anti-symmetrical operator in shallow water system, the units of the evolution variables must be unified, otherwise, the addition is not able to operate between different variables, this is the reason we take IAP transformation.

20 Improving the conservation property is not like improving accuracy of the model, the convergence rate of spatial discrete operator, and the accuracy of temporal integration scheme are not changed in our study. Indeed, the reductions of errors are not such significant, but the physical characteristics are more analogous to the real system. The differences are not obvious in short term simulation, but in long term simulation, the advantage of strict energy conservation scheme may be huge, this is intuitively showed by the numerical test in (Wang,1996), which we'd like to share with you in Response for specific comment

25 #2.

In the following content, we response your specific comments.

Response for specific comment #1:

Sections 3.1, 3.2. The notation  $\mathcal{L}F$ ,  $\mathcal{M}u$ ,  $\mathcal{N}\phi$  suggests that  $\mathcal{L}$ ,  $\mathcal{M}$ ,  $\mathcal{N}$  are linear functions of F, u, and - respectively. In fact they are all nonlinear functions. Moreover,  $\mathcal{M}$  and  $\mathcal{N}$  are actually functions of both u and  $\phi$ . These two sections are overelaborate and presented in a very confusing way. In several places it is not obvious what is assumed and what is claimed to be proved. All that is really needed is the fact that the energy at a point can be written as a squared quantity by making a certain

5 change of variables.

**Reply:**

Thank you for reminding, indeed, the derivation does not depend on the linear operation, but indeed the expression is not strict enough. The following expression is better

$$\begin{cases} \frac{\partial u}{\partial t} + \mathcal{M}(\phi, u) = 0\\ \frac{\partial \phi}{\partial t} + \mathcal{N}(\phi, u) = 0 \end{cases}$$

10 For simplify expression, we write  $\mathcal{M} = \mathcal{M}(\phi, u), \mathcal{N} = \mathcal{N}(\phi, u)$

$$\frac{\partial U}{\partial t} = \sqrt{\phi} \frac{\partial u}{\partial t} + \frac{u}{2\sqrt{\phi}} \frac{\partial \phi}{\partial t} = -\sqrt{\phi} \mathcal{M} - \frac{u}{2\sqrt{\phi}} \mathcal{N}$$
$$(\mathcal{L}(F), F) = -\left(\frac{\partial U}{\partial t}, U\right) - \left(\frac{\partial \phi}{\partial t}, \phi\right)$$
$$= \oint_{\Omega} -U \frac{\partial U}{\partial t} - \phi \frac{\partial \phi}{\partial t} ds$$
$$= \oint_{\Omega} -U \left(-\sqrt{\phi} \mathcal{M} - \frac{u}{2\sqrt{\phi}} \mathcal{N}\right) + \phi \mathcal{N} ds$$
$$= \oint_{\Omega} \phi u \cdot \mathcal{M} + \frac{|u|^2}{2} \mathcal{N} + \phi \mathcal{N} ds$$

15 =
$$\oint_{\Omega} \phi u \cdot \mathcal{M} + \frac{|u|^2}{2} \mathcal{N} + \phi \mathcal{N} ds$$

=  $(\mathcal{M}, \phi u) + (\mathcal{N}, E)$
= 0

All of the similar expressions are fixed in the new version manuscript.

About "the energy at a point can be written as a squared quantity by making a certain change of variables", this is what we are talking about in Section 3.1, the square conservation is equivalent to energy conservation in a continuous system.

Response for specific comment #2:

P1 line 27, also P2 lines 3-5. The opening sentence is too categorical. For a quantity like energy or potential enstrophy, in a numerical model the total is made up of resolved and unresolved contributions. Therefore, it is not obvious that conserving the

- 25 resolved contribution is necessary for a good solution; indeed, it may be necessary to dissipate the resolved contribution (e.g. to prevent 'spectral blocking'). One can argue for a conservative numerical method by saying that we want to parameterize any dissipation, not leave it to numerical errors, but the opposite argument can also be made. Such ideas are extensively discussed in the literature.
  - Reply:

20

Indeed, there are resolvable and unresolvable energy contributions. Since the model resolution is not able to reach a molecule level, the numerical model cannot resolve all the mass, there are resolvable and unresolvable mass contributions either, as widely known, it's hard to obtain a good result without total mass conservation. For total energy, the influence is not significant in short-term simulation, but the long-term simulation, without total energy conservation, often lead to a terrible result.

- 5 On the other hand, energy conservation is one of the intrinsic conservation properties of the spatial discrete operator in TRiSK shallow water dynamic core (Ringler et al, 2010, Section 3.7), however, this property is lost during temporal integration by using original Runge-Kutta scheme. The temporal integration scheme brings time truncation error into the model, rather than spatial discrete operator, which means that the temporal integration scheme makes the model loses one of the intrinsic properties which is provided by spatial discrete scheme.
- 10 Figure 3c in the manuscript, a detail is that the square conservation scheme strictly conserves energy, even though the steady geostrophic flow collapses, but the original TRiSK scheme cannot maintain the total energy after collapse, this is an obvious difference between the "Conserving energy within truncation error" and "Strictly conserving energy". The reason is that the square conservation scheme maintains the conservation properties of spatial discrete operators faithfully, but original temporal integration scheme does not.
- 15 An interesting example can be found in (Wang, 1996), the numerical test of the linear ODE

$$\begin{cases} \frac{dx}{dt} = -ay\\ \frac{dy}{dt} = bx \end{cases}$$

the true solution of the equation is an ellipse conform to  $bx^2 + ay^2 = c$  (*c* is a constant), after long term numerical simulation (after 108 steps) with original Runge-Kutta, the generalized energy tends to zero, and the solution tends to a single point (Fig. 2(a), Wang, 1996, as showing as follow), but the Runge-Kutta with square conservative property is able to maintain the generalized energy strictly conserved, and the solution is still a ellipse as initial time (Fig. 2(b), Wang, 1996).

20

Fig. 2. Results from the  $10^{4}$ -step integration (printing a result per  $10^{5}$  steps). (a) By the old Runge-Kutta scheme, (b) by the new Runge-Kutta scheme.

This class of Runge-Kutta scheme with square conservative property is exactly what we mentioned in the manuscript Section 3.3.

**5 Response for specific comment #3:**

P1 line 29, Figs. 3b, 5b, 7b, P19 line 17. On a spherical domain the vanishing of the global integral of vorticity is a purely kinematic identity. Provided the vorticity and its integral are calculated in a self-consistent way, the same result must hold in the discrete case (e.g. P10 line 20). Thus, conservation of the global integral of vorticity is a test only of the fact that the vorticity is calculated consistently; it says nothing about the properties of the numerical methods used to solve the equations.

10 Reply:

Total absolute vorticity conservation is one of the intrinsic properties of TRiSK shallow water dynamic core, in the manuscript, we are not trying to discuss the importance of absolute vorticity conservation, but to maintain the total energy conservation without breaking down the intrinsic properties of TRiSK, The figures and the demonstrations about the conservation of total absolute vorticity are here to prove that the square conservation scheme has no influence to the other conservation properties

15 of TRiSK shallow water dynamic core.

**Response for specific comment #4:**

P1 line 29. '...five basic physical conserved properties'. Actually potential enstrophy is just one member of an infinite family (so-called Casimirs).

20 Reply:

Thank you for reminding, indeed, potential enstrophy is one member of an infinite family, in the manuscript, we are not tending to discuss all of the invariants, the description about five basic physical conservative properties is based on (Wang, 2008).

Response for specific comment #5:

5 P3 line 23. This form of the equations is usually called 'vector-invariant' (as on P19 line 16).Reply:

Indeed, thank you very much, it has been fixed in the new version of manuscript

Response for specific comment #6:

10 P4. Equation (3) is inconsistent with the definition of the 2-norm on line 5. This paragraph seems to be mixing up point values and global integrals.

Reply:

Indeed, the total energy should be defined as follow

 $\oint_{\Omega} \epsilon \, ds = \oint_{\Omega} g \epsilon_{R10} \, ds = \oint_{\Omega} \phi K + \frac{1}{2} \phi^2 + \phi \phi_s \, ds = \|\phi K\| + \left\| \frac{1}{2} \phi^2 \right\| + \|\phi \phi_s\|$

15

Response for specific comment #7:

Figure 1. Note that the grid used need not be uniform and regular (as suggested by the figure).

Reply:

mesh.

Indeed, uniform and non-uniform grid do not influence the location of the variables and the structure of spatial discrete
operators are the same as well. The square conservation scheme is available on arbitrarily structured C-grids as the title of the manuscript. As shown in Fig.1, Fig.2, Fig.3, (Ringler et. al, 2010), the regular is clear to introduce the structure of the SCVT

Response for specific comment #8:

25 P5. Note that the sign convection for  $u_e$  is related to the direction of the unit normal -this is crucial to get everything to work out. Also crucial for energy conservation is that  $Q_e^{\perp}$  is constructed to satisfy the equation on P20 line 9.

**Reply:**

The description of indicator function  $n_{e,i}$  for identifying the direction of  $u_e$  can be found in the end of Section 2,  $n_{e,i}$  appears in all of the correlative derivations in the manuscript.

30 The spatial discrete operators, that we described in the manuscript, are the same as those in (Ringler et al., 2010), we construct the anti-symmetrical spatial discrete operator by using the original spatial discrete operator in TRiSK shallow water dynamic core, therefore, all of the properties, mentioned by (Ringler et. al,2010), are still applicable in the manuscript. Indeed, there are

two methods of calculating  $Q_e^{\perp}$  in (Ringler et al,2010), in our manuscript, the algorithm of calculating  $Q_e^{\perp}$  satisfies the condition to keep energy conservation, which is described in Section 3.7.2, Ringler et al. (2010).

Response for specific comment #9:

5 P5 line 22. '...new type of Runge-Kutta'. Not so new (1996). Reply:

It is not so new, we try not to modify the title of (Wang, 1996), "A Class of New Explicit Runge-Kutta Schemes", in the new version of manuscript, this expression is changed.

10 Response for specific comment #10:

P6 line 11. It would be helpful to give a reference for 'IAP transformation'.

**Reply:**

The earliest description about IAP transformation can be found in (Section 2, Zeng, 1987), and also cited by (Wang, 2004).

15 Response for specific comment #11:

P11 lines 22-23. The text does not make sense - it seems to be mixing up point values and global integrals. Reply:

Indeed, there is a mistake, the error measure function should be  $I(X^n) = \frac{S(X_i^n) - S(X_i^0)}{S(X_i^0)}$ , where  $S(X) = \frac{\sum_{i=1}^N X(i)A(i)}{\sum_{i=1}^N A(i)}$ ,  $X_i^n$  is the variable at the *n*th time point on the ith cell and  $X_i^0$  is the variable at the initial time, and A(i) is the area of the *i*th cell. This is

20 similar to (135)-(140), Williamson, 1992, but the coordinate is no longer latitude-longitude, in the quasi-uniformed grid, the weight is now area of each cell. For a simpler expression  $I(X^n) = \frac{\sum_{i=1}^{N} (X_i^n - X_i^0)A(i)}{\sum_{i=1}^{N} X_i^0A(i)}$ , the result is equivalent to the former expression.

Response for specific comment #12:

25 Section 5.2. The fact that the solid body rotation flow eventually breaks down, despite the conservation properties of the scheme, is intriguing. Could this be a manifestation of the 'Hollingsworth instability' (as discussed, for example, in Skamarock et al 2012)?

Reply:

We are not sure the connection between the collapse of steady geostrophic flow and 'Hollingsworth instability', but we found

30 another way to delay the collapse, we observed that once we site the cell centers on two poles, the poles are just like the sources of errors, so we tried to rotate the mesh, and did not let any cell center site on poles, the errors was much more smaller, and the collapse has delayed obviously. Therefore, in our opinion, the principal cause of collapse is not the conservation properties,

but maybe something like polar singularity, as we mentioned in the manuscript, maintaining the strict energy conservation just delays the collapse, this phenomenon needs further study.

Response for specific comment #13:

5 P19 line 26. Sign error? Line 30. What is  $h_e$ ?

Reply:

Thanks a lot.

P19 line 26 should be  $\left(U, \frac{\partial U}{\partial t}\right)_e + \left(\phi, \frac{\partial \phi}{\partial t}\right)_i = 0$ , and line 30 should be  $\left(U, \frac{\partial U}{\partial t}\right)_e = \sum_{e=1}^{nEdges} U_e \left(C_e \frac{\partial u_e}{\partial t} + \frac{u_e}{2C_e} \frac{\partial \phi_e}{\partial t}\right) A_e$ Even though, there is no influence to the result, these errors shouldn't be happened.

**10**

Response for specific comment #14:

References are not in alphabetical order.

**Reply:**

This problem is fixed in new version of manuscript. Thank you for reminding.

**15**

**Replys to Anonymous Referee #2**

We would like to thank you for the positive comments and constructive advices, which help us to make the manuscript more clearly and more persuasive. The responses for the comments are in following text.

20 Response for specific comment #1:

The necessity to conserve the resolved energy in numerical solutions to an energy conservation system is actually the same as that to conserve the resolved mass. To highlight the significance of constructing an energy conservation scheme for the TRiSK dynamic core, a clear explanation on the necessity should be provided in Section 1. Reply:

- 25 This is a good advice, energy conservation is an important property for the closed physical system, the shallow water system without any energy sink or source is one of the closed system, and the numerical model such as TRiSK shallow water dynamic core is a kind of approximation to the closed system, therefore, the basic integral invariants should be conserved, as we cited from (Arakawa, 1977), the maintenance of integral make the physics of the discrete model more analogous to the physics of the continuous atmosphere, and on the other hand make the errors less systematic. Another interesting example could be found
- 30 in (Wang, 1996), the numerical test of the linear ODE

$$\begin{cases} \frac{dx}{dt} = -ay\\ \frac{dy}{dt} = bx \end{cases}$$

the true solution of the equation is an ellipse conform to  $bx^2 + ay^2 = c$  (*c* is a constant), but after long term numerical simulation (after 108 steps) with original Runge-Kutta, the generalized energy tends to zero, and the solution tends to a single point(Fig.2, Wang, 1996). I think it's clearly to see the importance of keeping energy conservation. The references are packed in the zip file.

Response for specific comment #2:

Line 19/Page 2: CRK is improperly used as the abbreviation of "a new class of Runger-Kutta scheme", because the word "class" does not describe the main characteristics of this scheme. NRK is better.

10 Reply:

5

CRK stands for Conservative Runge-Kutta in my opinion, which means this kind of Runge-Kutta helps make the square conservation, it's just an abbreviation, but of course, the naming right belongs to the proposer of the scheme, Bin Wang. I use this abbreviation just to make the article concise.

15 Response for specific comment #3:

I wonder why the title of Section 2 is exactly the same as that of Section 1 (Lin 22/Page 3).

**Reply:**

Thank you for finding out the problem. The right title of Section 2 is "Introduction of TRiSK".

20 Response for specific comment #4:

The equality (3) (Line 4/Page 4) is not true, which missed the integration sign after the second equal mark.

**Reply:**

Indeed, the total energy should be defined as follow

 $\oint_{\Omega} \epsilon \, ds = \oint_{\Omega} g \epsilon_{R10} \, ds = \oint_{\Omega} \phi K + \frac{1}{2} \phi^2 + \phi \phi_s \, ds = \|\phi K\| + \left\| \frac{1}{2} \phi^2 \right\| + \|\phi \phi_s\|$

25

Response for specific comment #5:

The semi-discrete form of the shallow water equation set [Equations (4)-(5) on Lines 4-5/Page 5] should no longer be a partial differential equation set, but an ordinary differential equation set.

Reply:

30 We are trying to express the same discrete system as which in (Ringler, 2010) Eqs.(19)-(20), you're right, "semi-discrete form" should be modified to "discrete system".

Response for specific comment #6:

Line 6/Page 5: u and v are not the variables of Eqs.(4)-(5).

Reply:

5 You are right, u is the evolution variable for the equation, v does not appear in Eqs.(4) and (5).

Response for specific comment #7:

Line 20/Page 7: The equality is not true, because a negative sign is missed before (not sure, but there is only one equation) Reply:

10 Indeed, the derivation should be

$$\begin{cases} \frac{\partial u}{\partial t} + \mathcal{M}(\phi, u) = 0\\ \frac{\partial \phi}{\partial t} + \mathcal{N}(\phi, u) = 0 \end{cases},$$

For simplify expression, we write  $\mathcal{M} = \mathcal{M}(\phi, u), \mathcal{N} = \mathcal{N}(\phi, u)$

$$\frac{\partial U}{\partial t} = \sqrt{\phi} \frac{\partial u}{\partial t} + \frac{u}{2\sqrt{\phi}} \frac{\partial \phi}{\partial t} = -\sqrt{\phi} \mathcal{M} - \frac{u}{2\sqrt{\phi}} \mathcal{N}$$
$$(\mathcal{L}(F), F) = -\left(\frac{\partial U}{\partial t}, U\right) - \left(\frac{\partial \phi}{\partial t}, \phi\right)$$
$$15 = \oint_{\Omega} -U \frac{\partial U}{\partial t} - \phi \frac{\partial \phi}{\partial t} ds$$
$$= \oint_{\Omega} -U \left(-\sqrt{\phi} \mathcal{M} - \frac{u}{2\sqrt{\phi}} \mathcal{N}\right) + \phi \mathcal{N} ds$$
$$= \oint_{\Omega} \phi u \cdot \mathcal{M} + \frac{|u|^2}{2} \mathcal{N} + \phi \mathcal{N} ds$$
$$= (\mathcal{M}, \phi u) + (\mathcal{N}, E)$$
$$= 0$$

20 This problem does not influence the conclusion, thank you for checking the derivation meticulously.

Response for minor comment #8:

Line 10/Page 1: "The square conservation theory is widely used on latitude-longitude grids" -> "The square conservation law is maintained in the dynamic cores on latitude-longitude grids".

**25 Reply:**

The square conservation scheme is implemented in The Grid-point Atmospheric Model of IAP LASG(GAMIL), and the result of GAMIL was published in CMILP5, but your advice is good.

Response for minor comment #9 and #10:

30 9) Line 4/Page 2: "which is"  $\rightarrow$  "which are".

10) Line 26/Page 2: "polar problem" -> "polar instability" or "polar singularity".

**Reply:**

Thank you for finding out those mistakes, they are fixed in the 4th version of manuscript.

5

**List of relevant changes made in the manuscript**

- 1. Switch all of the CRK to NRK.
- 2. In Abstract, we fix some description of square conservation and add more introductions of two kinds of energy conservation scheme.
- 5 3. In Introduction
  - (1) Add more references to introduce the importance of energy conservation
  - (2) Expound the differences between conserving energy in time truncation-error and conserving energy in round-off error.
  - (3) Switch "polar problem" to "polar singularity".
- 10 4. In Section 2
  - (1) Switch "flux format" to "vector-invariant format"
  - (2) Fix the mistake of total energy expression as

$$\oint_{\Omega} \epsilon \, ds = \oint_{\Omega} g \epsilon_{R10} \, ds = \oint_{\Omega} \phi K + \frac{1}{2} \phi^2 + \phi \phi_s \, ds = \|\phi K\| + \left\|\frac{1}{2} \phi^2\right\| + \|\phi \phi_s\|$$

- (3) Add description for Figure 1.
- (4) Modify some other details.
- 5. In Section 3

15

- (1) Add references about IAP transformation.
- (2) Switch  $\mathcal{L}F$  to  $\mathcal{L}(F)$
- (3) Switch LF to L(F)
- 20 (4) Switch operators  $\mathcal{M}$  and  $\mathcal{N}$  to function  $\mathcal{M} = \mathcal{M}(\phi, u)$  and  $\mathcal{N} = \mathcal{N}(\phi, u)$
  - (5) Switch operators *M* and *N* to function  $M = M(\phi, u)$  and  $N = N(\phi, u)$
  - (6) Add description of  $\sqrt{\phi_e}$  and  $\phi_e$ .
  - (7) Correct the sign for Eq.(20)
  - (8) Modify some other details.
- 25 6. In Section 5
  - (1) Add description of the differences between conserving energy in time truncation-error and conserving energy in round-off error during entire temporal integration period.
  - (2) Correct  $I(X^n) = \frac{s(x_i^n) s(x_i^0)}{s(x_i^n)}$  to  $I(X^n) = \frac{s(x_i^n) s(x_i^0)}{s(x_i^0)}$
  - (3) Add the discussion about the benefits of implementing square conservation scheme in TRiSK in the end of this section.
  - 7. In Appendix A

(1) Correct

[revised manuscript text omitted]

---

## Author Response (AR2)

**Author's Response:**
First of all, we appreciate all the referees and editors, they gave us a lot of advices to improve this manuscript.

We add another author, Lihao Zhong, in the new version of manuscript, and the author Lijuan Hua has
5  changed her affiliation. There is no modification in the content of science.

[revised manuscript text omitted]